

# Addressing initialisation uncertainty for end-to-end ecosystem models: application to the Chatham Rise Atlantis model

Vidette L. McGregor[1], Elizabeth A. Fulton[2] and Matthew R. Dunn[1]

[1] Fisheries, National Institute of Water and Atmospheric Research Ltd, Wellington, New Zealand
[2] Oceans & Atmosphere, CSIRO, Hobart, TAS, Australia

## ABSTRACT

Ecosystem models require the specification of initial conditions, and these initial conditions have some level of uncertainty. It is important to allow for uncertainty when presenting model results, because it reduces the risk of errant or non-representative results. It is crucial that model results are presented as an envelope of what is likely, rather than presenting only one instance. We perturbed the initial conditions of the Chatham Rise Atlantis model and analysed the effect of this uncertainty on the model's dynamics by comparing the model outputs resulting from many initial condition perturbations. At the species group level, we found some species groups were more sensitive than others, with lower trophic level species groups generally more sensitive to perturbations of the initial conditions. We recommend testing for robust system dynamics by assessing the consistency of ecosystem indicators in response to fishing pressure under perturbed initial conditions. In any set of scenarios explored using complex end-to-end ecosystem models, we recommend that associated uncertainty analysis be included with perturbations of the initial conditions.

## INTRODUCTION

Atlantis models are end-to-end ecosystem models that can be used to model everything in an ecosystem from sunlight to market, including feed-back loops and flow-on effects (*Audzijonyte et al., 2017a*). There are many published Atlantis models (*Link, Fulton & Gamble, 2010*; *Ainsworth et al., 2011*; *Ainsworth, Schirripa & Morzaria-Luna, 2015*; *Fulton et al., 2011*; *Weijerman et al., 2014*; *Sturludottir et al., 2018*; *Ortega-Cisneros, Cochrane & Fulton, 2017*; *Porobic et al., 2019*; *McGregor et al., 2019b*), and sensitivities and uncertainties are beginning to feature in the more recently published work (*Ortega-Cisneros, Cochrane & Fulton, 2017*; *Sturludottir et al., 2018*, *Ortega-Cisneros et al., 2018*; *Hansen et al., 2019a*, *2019b*; *McGregor et al., 2019b*).

An important step that has not yet been addressed for Atlantis models, or any end-to-end ecosystem models, is to address initialisation uncertainty. In a review paper, *Payne et al. (2015)* found generally marine ecosystem models have not explicitly addressed

Corresponding author
Vidette L. McGregor,
vidette.mcgregor@niwa.co.nz

uncertainty of initialisation, and more recently, *Hansen et al. (2019a)* noted it is not
something that has been done for Atlantis models. *Payne et al. (2015)* speculated as to the
likely effects of initialisation uncertainty in end-to-end models such as Atlantis, noting
long-lived species might dampen effects, and short-lived species may amplify them. While
accounting for uncertainties of such complex models is no small task, it is possible to focus
on a sub-component of model dynamics rather than the entire model. Atlantis models
are particularly well suited for this approach as they are structured using components.
These components dynamically interact, and they define functionality for the physical,
biological, fisheries, harvest and economic components of the system (*Audzijonyte et al.,
2017b*). Rather than vary the initial conditions for thousands of spatially explicit
parameters for the entire ecosystem, we focus this study on the biological sub-component
and explore sensitivity to the initial conditions of species groups. These will have
flow-on effects to the whole system, but the analyses focus on the effects as they relate
to population dynamics of the modelled species.

Ecological theory has featured stability, chaos, the importance of initial conditions, and
how these relate, although definitions seem to vary. The most commonly known attribute
of chaotic dynamics is sensitivity to initial conditions, often referred to as the butterfly
effect. In a system with chaotic dynamics, a very small change in its initial state will result
in a different trajectory (*Baker, Baker & Gollub, 1996*). Stability often relates to variability,
persistence and robustness of a system (*Tilman, 1996*; *Ives, Gross & Klug, 1999*).
Multiple studies have investigated characteristics of an ecosystem that are linked with
stability, both from a theoretical perspective, and from observation. *May (1972)* showed
mathematically, that large complex systems with high levels of diversity are unstable.
However, there seem to be exceptions to this rule, as later studies have shown. *Roberts
(1974)* argued that most systems in practice appear to be more stable with more
connections—contrary to the mathematical analysis of *May (1972)*. *Roberts (1974)* showed
if only feasible solutions are included in the analyses, such that no species may have a
negative population, larger systems are actually more stable. *May (1975)* examined
non-linear difference equations with respect to chaotic, cyclic and stable biological
dynamics. Other aspects subsequently shown to increase stability of ecosystems include
negative pairwise correlations (*Tang & Allesina, 2014*), species dispersal (*Allesina & Tang,
2012*), modularity (subsets of closely connected components) (*Grilli, Rogers & Allesina,
2016*), predator-prey relationships (*Tregonning & Roberts, 1979*), a high proportion of
weak interactions (*Olsen et al., 2016*) and spatial structure (*Fulton, 2001*).

The Chatham Rise Atlantis model was selected for this study as it has been shown to
have sound dynamics with respect to our current knowledge of the system (*McGregor
et al., 2019b*), has been tested for sensitivity to its ocean forcing variables (*McGregor et al.,
2019b*) and spawning stock recruitment assumptions (*McGregor, Fulton & Dunn, 2019a*),
but is yet to be tested with respect to its initial conditions. The initial conditions for the
Chatham Rise Atlantis model were specified to reflect the ecosystem in its unfished, or
virgin state. We have varying levels of understanding of the components of this ecosystem
in its unfished state, and as such, there are varying levels of confidence around the
estimates for the initial conditions, with all components having some level of error.

This study presents an approach for addressing and exploring sensitivity to initial conditions of the biological sub-component of an end-to-end ecosystem model, with application to the Chatham Rise Atlantis model. We go beyond the question of whether the model is sensitive to its initial conditions, and analyse what features of the Chatham Rise ecosystem, and how we have modelled it, affect this answer. We discuss the likely impacts of our findings for future use of this ecosystem model. We highlight areas of potential future research with respect to model development, and to support decisions relating to the sustainable use of the Chatham Rise marine ecosystem resources.

## METHODS

The Chatham Rise Atlantis model is spatially defined using 24 dynamic polygons, and five water column depth layers. Species are modelled using 55 species groups, which include species of bacteria, detritus, phyto-plankton, invertebrates, fish, sharks, cetaceans and birds. Some species groups were modelled as biomass pools, and some with age-structure, using numbers-at-age and mean weight-at-age. For many of the species, we have estimates of biomass, growth rates, age of maturity, natural mortality, spatial distributions and diets, although some species have more knowledge gaps than others. *McGregor et al. (2019b)* characterised the species groups by keystoneness, responsiveness, and informance, and brought these together to highlight which data gaps are likely to influence model results. We can use these attributes to perturb the initial conditions in a meaningful way based on likely uncertainties, and they may add context when analysing the results. Keystoneness measures the effect changes in biomass of a species group has on the rest of the system; responsiveness measures how responsive a species group is to changes in biomass of other species groups within the system. Informance was a qualitative measure used to reflect both how well informed each species group was, and how well it performed in the model. It considered whether key dynamics such as growth, mortality rates, diets, and responses to fishing were all realistic based on current knowledge of these dynamics.

Sensitivity to initial conditions is characterised by two criteria in this study: (1) persisting variability between simulations, assessed at the species group level; (2) consistency of responses to fishing pressure, both at the species group level and at the system level. Variability of species biomass between simulations with perturbed initial conditions is important to consider as it may affect conclusions drawn from the model, and can provide error bounds around results. The consistency of responses to fishing pressure has also been considered as it is often the direction of a response that is of interest when exploring scenarios using ecosystem models. For example a conclusion that increasing fishing by 20% reduced diversity by 5–15% (with the range presented perhaps reflecting the initialisation uncertainty) is more informative than whether simulated biomasses of individual species groups varied across runs by less than 20%.

The analyses presented here were carried out in three main sections: (1) components of the modelled system were characterised with respect to attributes that may affect sensitivity to initial conditions; (2) the initial conditions were perturbed, and the resulting model simulations were compared; (3) correlations between component attributes and
responses to perturbations of the initial conditions were analysed, thus linking the first two sections.

The models were run on a Linux cluster using Moab job scheduler, and used Atlantis version 6262M (trunk). We created a new initialisation.nc file for each set of perturbed initial conditions, and these were created using R (version 3.6.1) to read in the base initialisation file, scale the values, then create the new initialisation file. R was also used to create run files that interacted with Moab to schedule the model runs. Scripts used for this work are available on GitHub (*McGregor, 2018*) (https://github.com/mcgregorv/CRAM_chaos). The R scripts were all developed in a Windows environment.

## Varying initial conditions

We varied the initial conditions for the number-at-age variables of age-structured species groups, and the biomass of biomass-pool species groups (Tables 1 and 2). For the biomass-pool species groups, biomass is the only option to perturb. Age-structured species groups could have errors in the specification of numbers-at-age and/or size-at-age, both of which affect the biomass-at-age. In a stock assessment model, size-at-age (or growth rates) are generally the same with respect to time, whereas there is more likely a difference in numbers with respect to time (especially before fishing compared to after fishing). Hence, a different virgin biomass in a stock assessment model would generally be made up of a different number of fish, rather than the same number of fish but a different size. To align with this, we perturbed numbers rather than size for the age-structured species groups. The resulting number of variables to perturb was 361, of which 341 were numbers-at-age (number of age-classes for a species group ranged from 2 to 10, with 10 being most common), 18 were the nitrogen content of biomass-pool species groups, and two were the silicate content of biomass-pool species (diatoms and microphytobenthos). The numbers-at-age of all age-classes for a given species group were scaled by the same amount for each simulation, such that the proportions-at-age were preserved. Again, this aligns most closely with the structure of a population's virgin state in a stock assessment. The assumed $M$ (instantaneous natural mortality) is preserved through the proportions-at-age based on an exponential decay curve, and the total numbers are what change when varying the initial conditions. Age-structured species groups were modelled with between 2 and 10 age-classes, and were perturbed by applying one scalar for all age-classes of a given species group. This significantly reduced the number of scalars required to 57, of which 37 were for scaling numbers-at-age for age-structured species groups.

Initially, we perturbed all species group initial conditions using the same scalar for all variables within each model run. The scalars we used were 0.5, 0.8, 0.9, 0.95, 1.05, 1.1, 1.2, 1.5. These scalars were chosen to cover a range from slight (±5%) to extreme (±50%) errors in the initial conditions.

As shifting initial conditions by the same amount may not give an indication as to how robust or sensitive the model is to mis-specification of the initial conditions where changes could vary in direction and magnitude, we next simulated multiple model runs, with the initial conditions scaled with some random variability. We scaled the initial conditions

**Table 1 List of age-structured species groups for CRAM (*McGregor et al., 2019b*).** 'Keystone' is the keystone ranking from *McGregor et al. (2019b)*, where 1 is the highest. 'Informance' is the informance rating from *McGregor et al. (2019b)* where 1 is the highest ('No data gaps, performed well, abundance index available') through to 4 which is the lowest ('Poorly specified'). Ben: benthic; Dem: demersal; invert: in-vertivore; pisc: piscivore; Invert comm: commercial invertebrates; herb: herbivore; scav: scavenger.

| Species group | Main species | Keystone | Informance |
|---|---|---|---|
| Arrow squid | Arrow squid | 19 | 3 |
| Baleen whales | Southern right whales (*Eubalaena australis*) | 25 | 4 |
| Basketwork eel | Basketwork eels (*Diastobranchus capensis*) | 18 | 3 |
| Baxters dogfish | Baxter's dogfish (*Etmopterus baxteri*) | 28 | 2 |
| Ben fish deep | Four-rayed rattail (*Coryphaenoides subserrulatus*) | 35 | 2 |
| Ben fish shal | Oblique banded rattail (*Coelorinchus aspercephalus*) | 5 | 1 |
| Black oreo | Black oreo (*Allocyttus niger*) | 26 | 2 |
| Bollons rattail | Bollons' rattail (*Caelorinchus bollonsi*) | 16 | 1 |
| Cephalopod other | Squid & octopus | 14 | 2 |
| Cetacean other | Primarily sperm & pilot whales & dolphins | 9 | 4 |
| Dem fish pisc | Giant stargazer (*Kathetostoma giganteum*) | 31 | 3 |
| Elasmobranch invert | Primarily skates & dogfish | 32 | 1 |
| Elasmobranch pisc | Primarily semi-pelagic sharks | 17 | 2 |
| Epiben fish deep | Spiky oreo (*Neocyttus rhomboidalis*) | 22 | 2 |
| Epiben fish shal | Common roughy (*Hoplostethus atlanticus*) | 8 | 2 |
| Ghost shark | Dark ghost shark (*Hydrolagus novaezealandiae*) | 30 | 2 |
| Hake | Hake (*Merlucciidae australis*) | 10 | 1 |
| Hoki | Hoki (*Macruronus novaezelandiae*) | 1 | 1 |
| Invert comm herb | Paua (*Haliotidae*) & kina (*Evechinus chloroticus*) | 21 | 2 |
| Invert comm scav | Primarily scampi & crabs | 33 | 2 |
| Javelinfish | Javelinfish (*Coelorinchus australis*) | 27 | 1 |
| Ling | Ling (*Genypterus blacodes*) | 11 | 2 |
| Lookdown dory | Lookdown dory (*Cyttus traversi*) | 20 | 1 |
| Mackerels | Slender jack mackerel (*Trachurus murphyi*) | 15 | 2 |
| Orange roughy | Orange roughy (*Hoplostethus atlanticus*) | 2 | 1 |
| Pelagic fish lge | Southern bluefin tuna (*Thunnus thynnus*) | 29 | 2 |
| Pelagic fish med | Barracouta (*Thyrsites atun*) | 7 | 3 |
| Pelagic fish sml | Myctophids (*Myctophidae*) | 4 | 3 |
| Pinniped | NZ fur seal (*Arctocephalus forsteri*) | 36 | 4 |
| Reef fish | Blue cod (*Parapercis colias*) | 24 | 2 |
| Rock lobster | Rock lobster (*Jasus edwardsii*) | 37 | 3 |
| Seabird | Seabirds & shorebirds | 6 | 4 |
| Seaperch | Seaperch (*Helicolenus* spp.) | 34 | 2 |
| Shovelnosed dogfish | Shovelnosed dogfish (*Deania calcea*) | 12 | 2 |
| Smooth oreo | Smooth oreo (*Pseudocyttus maculatus*) | 23 | 2 |
| Spiny dogfish | Spiny dogfish (*Squalus acanthias*) | 3 | 2 |
| Warehou | Silver, white & blue warehou (*Seriolella* spp.) | 13 | 3 |

**Table 2 List of species groups modelled as biomass-pools for CRAM (*McGregor et al., 2019b*).** Zoo: zooplankton.

| Species group | Description |
| --- | --- |
| Benthic Carniv | Benthic carnivores |
| Carniv Zoo | Planktonic animals (size 2–20 cm) |
| Carrion | Dead and decaying flesh |
| Deposit Feeder | Detritivores and benthic grazers |
| Diatoms | Diatoms (large phytoplankton) |
| DinoFlag | Dinoflagellates |
| Filter Other | Non-commercial benthic filter feeders |
| Gelat Zoo | Salps, ctenophores, jellyfish |
| Labile detritus | Organic matter that decomposes at a fast rate |
| Macroalgae | Macroalgae |
| Meiobenth | Benthic organisms (size 0.1–1 mm) |
| MesoZoo | Planktonic animals (size 0.2–20 mm) |
| Microphytobenthos | Unicellular benthic algae |
| MicroZoo | Heterotrophic plankton (size 20–200 µm) |
| Pelagic bacteria | Pelagic bacteria |
| Pico-phytoplankton | Small phytoplankton |
| Refractory detritus | Organic matter that decomposes at a slow rate |
| Sediment bacteria | Sediment bacteria |

of each variable, sampling the scalar for each from a normal distribution, $N(0, \sigma)$ with $\sigma$ chosen based on how large we assumed a plausible change could be.

In total, we ran three sets of simulations, and repeated each set with and without fishing.

Set 1: All up or down. All species group initial conditions were scaled (numbers for age-structured, biomass for biomass-pool) with the same scalar for each run;

scalars $\in$ {0.5, 0.8, 0.9, 0.95, 1.05, 1.1, 1.2, 1.5}

Set 2: High uncertainty. All initial conditions were scaled (numbers for age-structured, biomass for biomass-pool), with the scalars sampled from normal distributions with $\mu = 0$ and $\sigma$ set based on the informance ratings defined in *McGregor et al. (2019b)* (Fig. 1). Biomass-pool species groups were assumed poorly specified as these were not ranked in *McGregor et al. (2019b)*.

Set 3: High keystone species. These runs only scaled the initial conditions of species groups likely to be most influential on the system. The species groups that ranked in the top 10 for keystoneness in *McGregor et al. (2019b)*, and all biomass pool species groups were scaled using normally distributed scalars sampled with $\mu = 0$ and $\sigma = 0.25$, giving 95% confidence intervals of $\approx \pm 0.5$. All other species groups were unchanged (Fig. 2).

## Ecosystem indicators

To help our understanding of the sensitivity of the whole model to perturbations of the initial conditions, we analysed ecosystem indicators from all simulations. We calculated a subset of the ecosystem indicators analysed for the base model in *McGregor, Fulton & Dunn (2019a)*
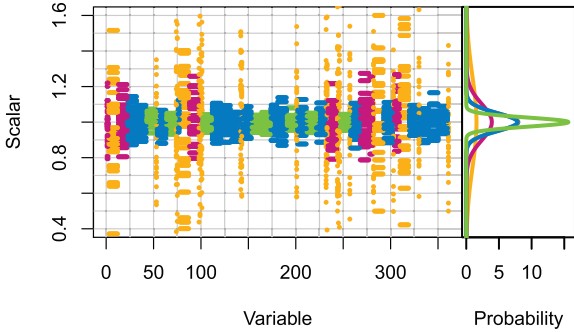

**Figure 1** Set 2 scalars used to perturb initial conditions, with scalars sampled from Gaussian distributions with μ = 0 and σ one of (0.025, 0.05, 0.1, 0.25) based on informance levels 1–4 respectively where (1) 'No data gaps, performed well, abundance index available' (green); (2) 'Slight data gaps and/or poor performance' (blue); (3) 'Some (more substantive) data gaps and/or poor performance' (magenta); (4) 'Poorly specified' (gold); (defined in *McGregor et al. (2019b)* and for reference in Table 1).

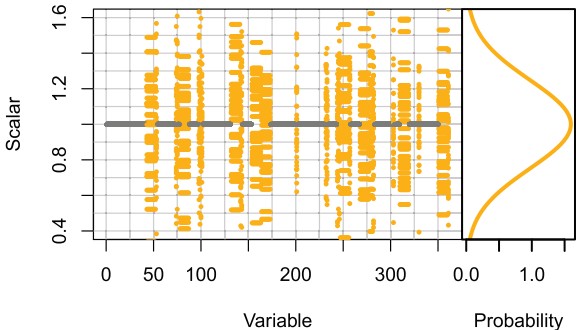

**Figure 2** Set 3 scalars used to perturb initial conditions for high keystone and biomass-pool species groups. Scalars sampled from the Gaussian distribution with μ = 0 and σ = 0.25.

(Table 3) at each timestep for all model simulations. Mean trophic level, diversity and the ratio of pelagic to total biomass were chosen as they responded to fishing scenarios for the Chatham Rise Atlantis model (*McGregor, Fulton & Dunn, 2019a*), but didn't require catch for the calculation (so we could apply them to model runs here with or without fishing included). We analysed the results for consistencies or discrepancies in shifts of the ecosystem reflected through these indicators, with particular focus on the response of the system when heavy fishing became established during the mid-1970s.

## Characterising the species groups

If certain species groups appear to be more stable than others, we wanted to be in a position to investigate whether the more stable species groups have shared characteristics—for example are there links between sensitivities to changes in the initial conditions and how connected each species group is in the system, how abundant they are, how long they live, or some combination of these.
**Table 3 Key ecosystem indicators evaluated for responses to perturbing the initial conditions.**

| Indicator | References |
| --- | --- |
| Mean trophic level | *Pauly & Watson (2005)* and *Shin et al. (2018)* |
| Diversity (modified Kempton's Q) | *Ainsworth & Pitcher (2006)* |
| Biomass of pelagic fishes/biomass total | *Link (2005)* |

We characterised the species groups based on the base model presented in *McGregor et al. (2019b)* so we could test for links between these attributes of the species groups and sensitivities to changes in the initial conditions. We considered keystoneness, trophic level, biomass, animal size, lifespan, additional (background) mortality, number of trophic connections, and proportion of most dominant ('top') prey. All but the final three of these indices were available from *McGregor et al. (2019b)*. The proportion of diet made up by most dominant prey, number of trophic connections, and the proportion of natural mortality that was made up of additional mortality were calculated for this study using R version 3.4.3.

### Proportion of top prey

For each species group, we calculated the contribution to a predators diet from the single most dominant prey in their diet, as a proportion of biomass consumed, using averaged diets from the base model. The intent was to classify the extent to which each species group was eating as a specialist or generalist as they are modelled. It is possible for a predator to perform in the model as more of a specialist due to aggregation of species into groups—they could predate on several prey species that are modelled in the same species group. For each species group, we summed the prey eaten over the entire model region and all modelled years 1900–2015, then selected the largest proportion. Due to this averaging, a predator switching between two prey (for example) would appear similar to a predator consistently consuming those two prey. Hence, a predator would need to consistently predate on a given prey species group to be considered a specialist feeder for this metric.

### Number of trophic connections

Trophic connections were calculated as 'primary connections' (predators and prey of the species group), 'secondary connections' (predators and prey of the predators and prey of the species group) and 'tertiary connections' (similarly) (illustration in Fig. 3). As sometimes a predator may eat a very small, negligible amount of a prey, we included a cut-off at 1%, such that a prey or predator was not included in the connections count if they made up less than 1% of the total prey consumed or predation pressure applied, respectively.

### Additional natural mortality

There is the option in Atlantis to apply additional natural mortality either as a quadratic term, which is density dependent, or as a linear term (*Audzijonyte et al., 2017a*). The balance between additional natural mortality and mortality coming from dynamics within the model may affect the model's stability. Higher levels of additional mortality

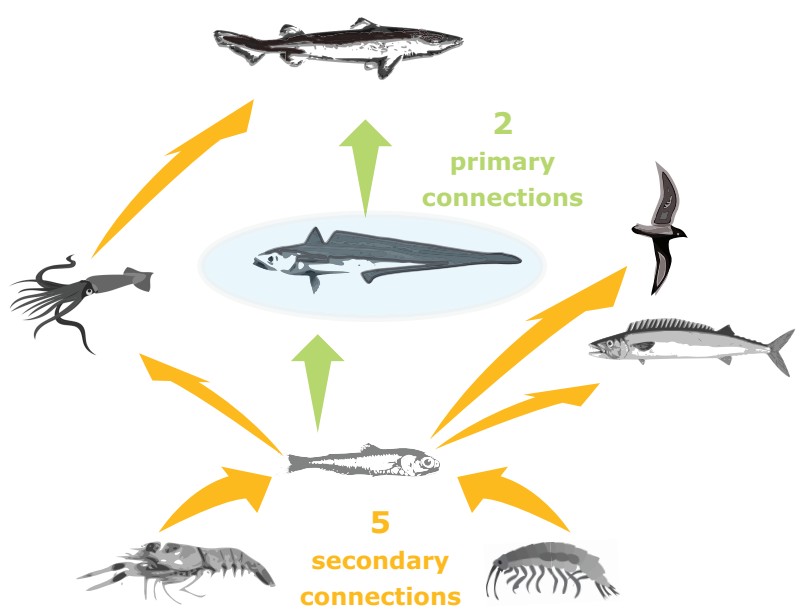

**Figure 3 Illustration of primary and secondary tertiary trophic level connections, where the species group in the centre (shaded blue) has two primary connections, and five secondary connections.**

reduce the strength of connections in the model, with 100% additional mortality effectively resulting in parallel single species models. Additional mortality was required for some species groups in the Chatham Rise Atlantis model that did not suffer sufficient natural mortality through predation, starvation or disease in the model to match estimates of mortality from the literature. For all age-structured species groups in this model, linear rather than quadratic mortality was applied as this is a close approximation to instantaneous natural mortality ($M$). When $M$ is small, as it is when applied at small time-steps, $e^{-M}$ can be approximated by $1 - x$ using the first two terms of its Taylor series expansion. Hence, if we take $N_t$ to be the number of individuals at timestep $t$ and $N_{t+\delta}$ to be the number at timestep $t + \delta$ where $\delta$ is small we get

$$N_{t+\delta} = N_t e^{-M_\delta} \approx N_t(1 - M_\delta) \tag{1}$$

As linear mortality, $m_L$, is applied at every timestep (12 h for this model), we can use $M_\delta$ to approximate $m_L$. This is, however, complicated by a temperature effect which is applied to $m_L$ in Atlantis. Additional mortality in Atlantis are assumed related to metabolic rates, and hence are temperature dependent. The temperature effect is applied as a scalar ($T_{corr}$) calculated as a function of the current water temperature ($T$) (in a given cell at a given time) relative to a base temperature, set at 15 °C (Eq. (2)).

$$T_{corr} = 2^{(T-15)/10} \tag{2}$$

As temperature varies spatially and temporally, so does the scaled $m_L$. We calculated the additional applied mortality for each species group based on their spatial distribution, $m_L$ values, and temperature corrections, using the median, upper and lower quartile, and

**Table 4 Explanatory variables offered to explain effects of perturbing the initial conditions, and whether these were defined specifically for age-structured species groups or for all species groups.** Explanatory variables were analysed using the base model presented in *McGregor et al. (2019b)*.

| Variable | Description | All species groups |
|---|---|:---:|
| (1) Informance | Rating of how well informed and how well it performed in the base model | |
| (2) TL | Trophic level | ✓ |
| (3) Keystone | Keystone ranking | |
| (4) Response | Responsive ranking | |
| (5) NumL1cons | Number of primary trophic connections | ✓ |
| (6) Lifespan | Approximate expected lifespan in years | ✓ |
| (7) propAdM | Proportion of adult natural mortality that is estimated to have come from additional (background) mortality | |
| (8) propJuvM | Proportion of juvenile natural mortality that is estimated to have come from additional (background) mortality | |
| (9) B0 | Unfished biomass | ✓ |
| (10) PropByTopPrey | Proportion of diet made up by the most dominant prey group | |
| (11) Linf | Expected maximum length (cm) | |
| (12) ChaosAlt | Method used to perturb initial conditions (set 1, 2 or 3) and whether fishing was turned on or not | ✓ |

95% confidence intervals for the applied additional mortality to reflect the variability of temperature spatially and temporally. These were calculated for both juveniles and adults as $m_L$ and spatial distributions were defined separately for these life stages.

Total realised mortality rates were estimated from the model by fitting an exponential decay curve to the proportions-at-age. By running the model with no fishing, the realised mortality consisted entirely of natural mortality, including sources within the model such as predation, as well as additional mortality from $m_L$. We then compared the total realised natural mortality with the range of additional mortality to estimate what proportion of natural mortality was coming from dynamics within the model, and what proportion was forced. We produced a weighted average for each species group that combined the proportions for adults and juveniles, weighted by the numbers of adults and juveniles respectively.

## Modelling stability

We analysed variability as a result of perturbing the initial conditions for each species group by calculating the coefficient of variation (CV) of biomass between the runs. A species group with high variability between model simulations relative to the mean biomass from all simulations would have a high CV, and hence be considered highly sensitive to perturbations to the initial conditions.

We analysed the effects of perturbing the initial conditions by fitting a GLM (Generalised Linear Model) to the CV for the biomass of each species group across model runs. We used Atlantis model outputs following a 35-year burn-in period, to match the burn-in used in *McGregor et al. (2019b)*. Variables from characterising the species

**Table 5 ChaosAlt definitions for perturbing the initial conditions, and including fishing in the model or not.** ChaosAlt was offered as an ex-planatory variable to the GLMs.

| ChaosAlt | Description | Included fishing |
|---|---|---|
| A | All up or down | |
| B | All up or down | ✓ |
| C | Based on uncertainty | |
| D | Based on uncertainty | ✓ |
| E | Based on keystoneness | |
| F | Based on keystoneness | ✓ |

groups (Table 4) using the base model were offered as possible explanatory variables, using a step-wise selection algorithm, with each iteration selecting the variable (or pair of interaction variables) that explained the largest proportion of the null deviance. This process was repeated until the additional deviance explained was less than 10%. This cut-off value was selected to limit the number of explanatory variables selected, while retaining most of the explained null deviance. We initially explored untransformed, and log (base 10) and cubed root transformations of the response variable (CV), with all modelled using the Gaussian distribution. The analyses presented here used the cubed root transformation as we found this produced greater homogeneity of residuals with respect to the fitted values.

We could not model the biomass-pool species group CVs with respect to all attributes, as some attributes had not been analysed for biomass-pool groups (e.g. Keystone and Response), and some attributes relate to individuals, such as maximum size and instantaneous mortality. Hence, we fitted three versions of the GLM: (1) limited the species groups included in the analyses to species with age-structure in order to consider the full list of explanatory variables; (2) retained all species groups, but limited the explanatory variables offered to those that relate to all species groups; (3) limited the species groups to biomass-pool species groups, with the limited explanatory variables offered. Table 4 gives the full list of explanatory variables offered for biomass-pool (BP), age-structured (AS) and all-species (ALL) versions of the model. All possible paired interaction terms were also offered. PropByTopPrey was dropped from BP models as nearly half (8/17) of the biomass-pool species groups were not predators, and this variable only applies to predators.

We fitted the GLM to model outputs for each year (1900–2015) to test for temporal shifts in the effects (a separate GLM was fitted at each year). To allow for influence from the method of perturbing the initial conditions (all up or down, based on keystoneness, or based on uncertainty), we included this ('ChaosAlt') as a potential explanatory variable. We also explored splitting out the fished model runs from the unfished, or including this within ChaosAlt (Table 5).

We fitted a summary GLM for each version (ALL, AS, BP species; with/without fishing included), using a subset of the years simulated by the models, where the explanatory variables selected for models fitted at each timestep were roughly consistent. We used these

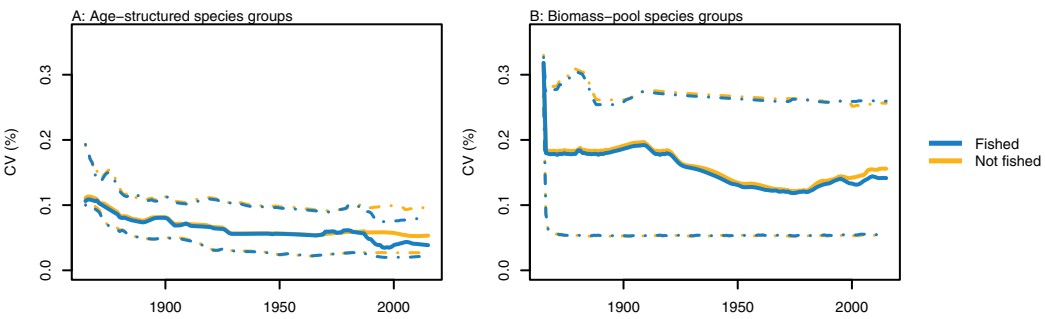

**Figure 4 CVs from model runs with perturbed initial conditions.** (A) Age-structured species groups, (B) Biomass-pool species groups. Median (solid lines) and upper and lower quartiles (dot-dashed lines) for CVs from fished model runs (blue lines) and un-fished model runs (orange lines).

summary models to explore the effects of the selected explanatory variables on between-run CVs. We analysed the residuals to check for trends or biases in the model fits, and present these as well as the effects of selected explanatory variables.

## RESULTS

### Variability from initial conditions

Variability between simulations with perturbed initial conditions remained high for some species groups, while others converged to almost identical outputs. Generally, the biomass-pool species groups were more likely to have persistent high CVs between model runs (Fig. 4). Fishing sometimes reduced the between-model CVs for age-structured species groups, such as for hoki, but the effects of fishing were not apparent in any biomass-pool species groups—in biomass trajectories or CVs between model runs (Fig. 5 for two examples; Appendix A for the full set of figures). Responses of age-structured species groups to fishing were generally consistent across model runs. This included direct effects of fishing on a species (such as hoki, hake, orange roughy and ling), and predation-release responses (such as cephalopods and pelagic fish). Exceptions were invert comm herb (primarily paua and kina), invert comm scav (primarily scampi), dem fish pisc (primarily giant stargazer) and seaperch, which all gave varied responses with fishing included in the model.

### Ecological indicators

Ecological indicators demonstrated variability from the perturbed initial conditions that generally persisted throughout the model simulations. However, the responses to heavy fishing from the mid-1970s were consistent across runs, with a decline in mean trophic level, a slight increase in diversity, and an increase in the ratio of pelagic biomass over total biomass (Fig. 6). There was a slight decline in mean trophic level from 1900 to 2015 in some of the unfished models, although the decline was approximately 0.02 of a trophic level over 100 years, so rather small. The ecological indicators present with more overlap between simulations than the individual species biomass outputs (Figs. 5 and 6).

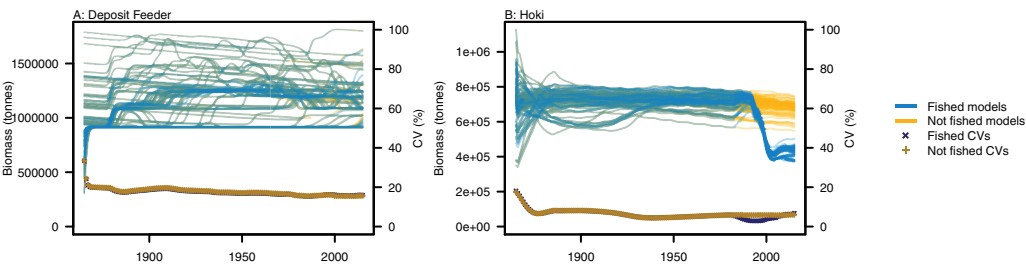

**Figure 5 Biomass trajectories for (A) deposit feeders and (B) hoki from model runs with perturbed initial conditions.** Models with fishing included (blue lines) and no fishing (orange lines), with CVs from across the model runs by time from fished models (midnight blue crosses) and un-fished models (dark orange pluses).

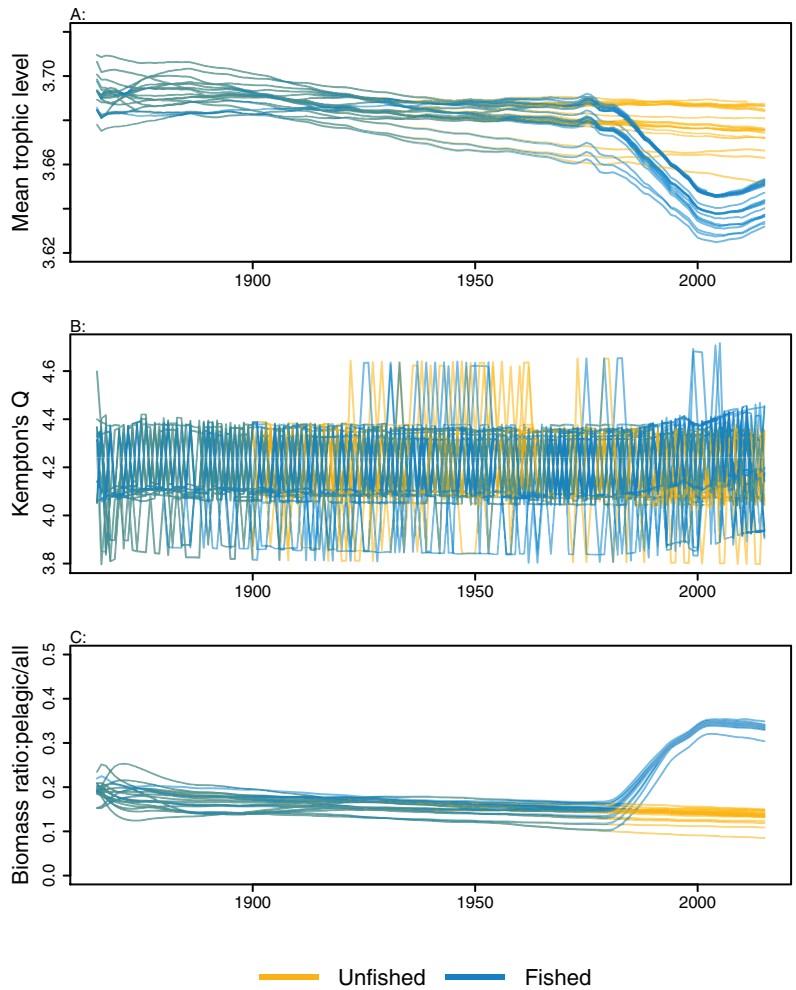

**Figure 6 Ecological indicators, mean trophic level of age-structured species groups, Kempton's Q, and biomass ratio of pelagic fishes/all age-structured species groups calculated from model simulations with perturbed initial conditions.** Mean trophic level of age-structured species groups (A), Kempton's Q (B), and biomass ratio of pelagic fishes/all age-structured species groups (C). Fishing included (blue lines), and no-fishing included (orange lines).

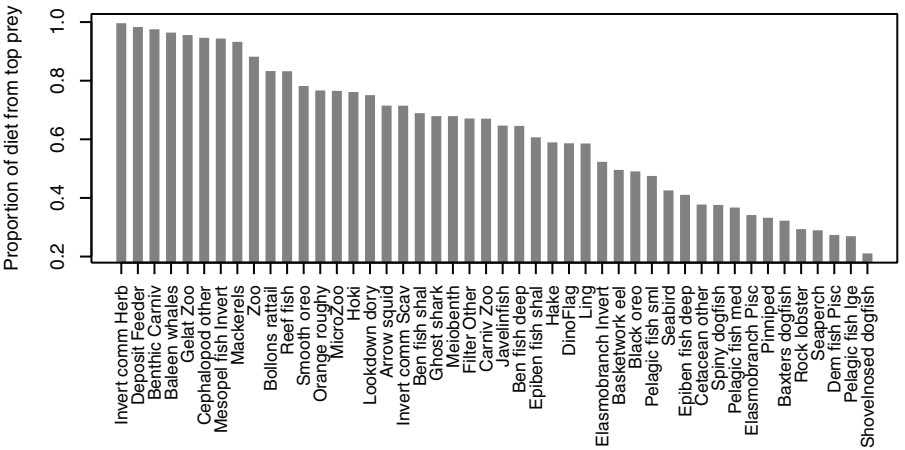

**Figure 7 Proportion of diet made up by top prey from the Chatham Rise Atlantis model (base) 1900–2015 model outputs.**

## Characterising the species groups

### Proportion of top prey

Some diets consisted almost entirely of one species group, but many others did not have a dominant species with the 'top' prey making up less than 50% of the diet, and there was quite an even spread in between, with top prey making up around 50–70% of many diets (Fig. 7). Not all species groups predate, which is why some species groups (such as sediment bacteria, macroalgae) do not have a highest proportion of prey and are not included in the plot.

### Number of trophic connections

The number of primary connections ranged from 1 through to 30, and with fairly even spread in between (Fig. 8). Most species groups were almost fully connected by the third level, and all species groups had at least 43 tertiary connections, of the 55 available species groups. Many of the species groups had more than 20 secondary connections, and those with fewer secondary connections generally had fewer primary connections. The number of secondary and tertiary connections are unlikely to be informative for stability between runs as there is little contrast.

### Additional natural mortality

The proportion of natural mortality forced with additional mortality through the $m_L$ term ranged from just over 0.8 for spiny dogfish down to zero for several species (Fig. 9). While baleen whales, cetacean other, pinnipeds and seabirds all have zero additional mortality through $m_L$, this does not mean their populations are entirely constrained due to mortality within the model, as these groups all migrate out of the model and their populations are restrained on re-entry into the model domain. Pelagic fish small (primarily myctophids), arrow squid, cephalopods other, and invert comm scav (primarily scampi) have all their natural mortality from sources such as predation within the model. Just over half (19/37) of the age-structured species groups had more than 80% of natural

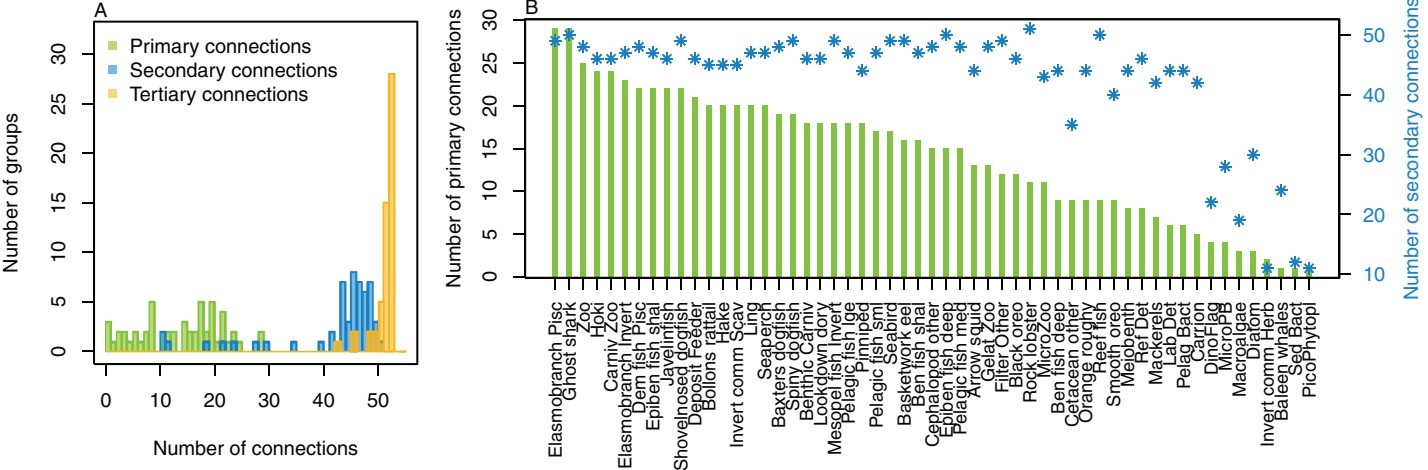

**Figure 8 Number of trophic level connections by species group for the Chatham Rise Atlantis model (base) 1900–2015 model outputs.** (A) Number of groups (frequency counts) by primary connections (green bars), secondary connections (blue bars) and tertiary connections (orange bars); (B) Number of primary connections by species group (green bars), and number of secondary connections by species group (blue asterisks, and using right-hand axis).

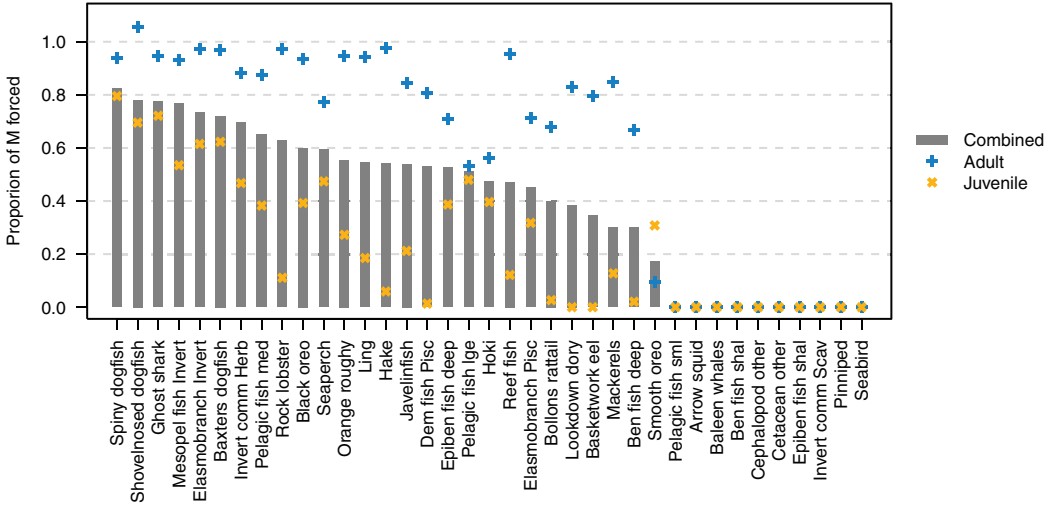

**Figure 9 Proportion of natural mortality (M) forced as additional mortality by species group from the Chatham Rise Atlantis model (base) 1900–2015 model outputs.** Additional mortality rates are not exact due to approximations of the temperature effects, hence the potential for a proportion of mortality that was forced to exceed 1.

mortality forced as adults. Most age-structured species groups (31/37) had less than 50% of natural mortality forced for juveniles.

## Modelling stability

### GLMs fitted at each timestep

The models fitted at each year with all species groups combined (ALL) selected the interaction term ChaosAlt (the way in which the initial conditions were perturbed) and trophic level, and explained 43–53% of the null deviance ($r^2$) (Table 6). The ChaosAlt: trophic level interaction term was also the most important explanatory variable for

**Table 6 Ranges of $r^2$ values for selected explanatory variables from GLMs fitted to between-run CVs at each year 1900–2015 for each subset of species groups (BP: Biomass Pool species groups; AS: Age-structured species groups, and ALL: all species groups).** Each column reflects three subsets of model runs (without fishing; with fishing; both with and with-out fishing). The range in $r^2$ values reflects the year variability as well as whether or not fishing was included in the model. The numbers in brackets are the number of years (out of the 116 years for which GLMs were fitted) the explanatory variable was selected, with the range re-flecting the variability from whether or not fishing was included in the model. The AS results are also presented in Fig. 10.

| Explanatory variable | BP | AS | ALL |
|---|---|---|---|
| ChaosAlt:Linf | | 0–0.01 (4–6) | |
| ChaosAlt:NumL1cons | | 0.04–0.06 (38–59) | |
| ChaosAlt:TL | 0.4–0.41 (103–107) | 0.31–0.31 (102–105) | 0.43–0.53 (116–116) |
| Keystone:propAdM | | 0.01–0.01 (6–7) | |
| Lifespan:Linf | | 0–0.03 (3–25) | |
| NumL1cons:B0 | 0.16–0.17 (103–107) | | |
| propAdM:Linf | | 0–0.04 (4–29) | |
| TL:B0 | 0.04–0.05 (9–13) | | |
| TL:Informance | | 0.01–0.02 (7–10) | |
| TL:NumL1cons | 0.02–0.02 (9–13) | | |
| TL:propAdM | | 0.01–0.01 (4–4) | |

biomass-pool (BP) only species group models, explaining 40–41% of the null deviance, and age-structured (AS) only species group models explaining 31% of the null deviance (Table 6). For all models, ChaosAlt:trophic level was selected as an explanatory variable for most years (at least 102 out of 116) (Table 6). BP models consistently selected a second term, which was generally the interaction of the number of primary trophic connections and virgin biomass ($B_0$) and explained an additional 16–17% of the null deviance. The AS models had different explanatory variables selected at different timesteps, and these were also influenced by whether fishing was included in the models (Table 6; Fig. 10). The interaction between ChaosAlt and the number of primary trophic connections was the most consistently selected second explanatory variable for AS models (Table 6; Fig. 10). The AS models seemed to have a shift at around 1910, and explanatory variables selected prior to 1910 did not include ChaosAlt, but trophic level and informance were important (Fig. 10).

### Final GLMs

The GLMs fitted to all data from 1910 to 2015 selected similar explanatory variables to the GLMs fitted at each timestep (Table 7). The interaction term ChaosAlt:TL was selected first for all models, and was the only term selected for the ALL model (all-species, with fishing and non-fishing runs included). The BP (biomass-pool) only species model also selected the interaction term NumL1cons:$B_0$. The AS (age-structured) species only model selected interaction ChaosAlt:NumL1cons whether fishing was included or not, and a third term, interaction NumL1cons:Informance was selected for the unfished AS model.

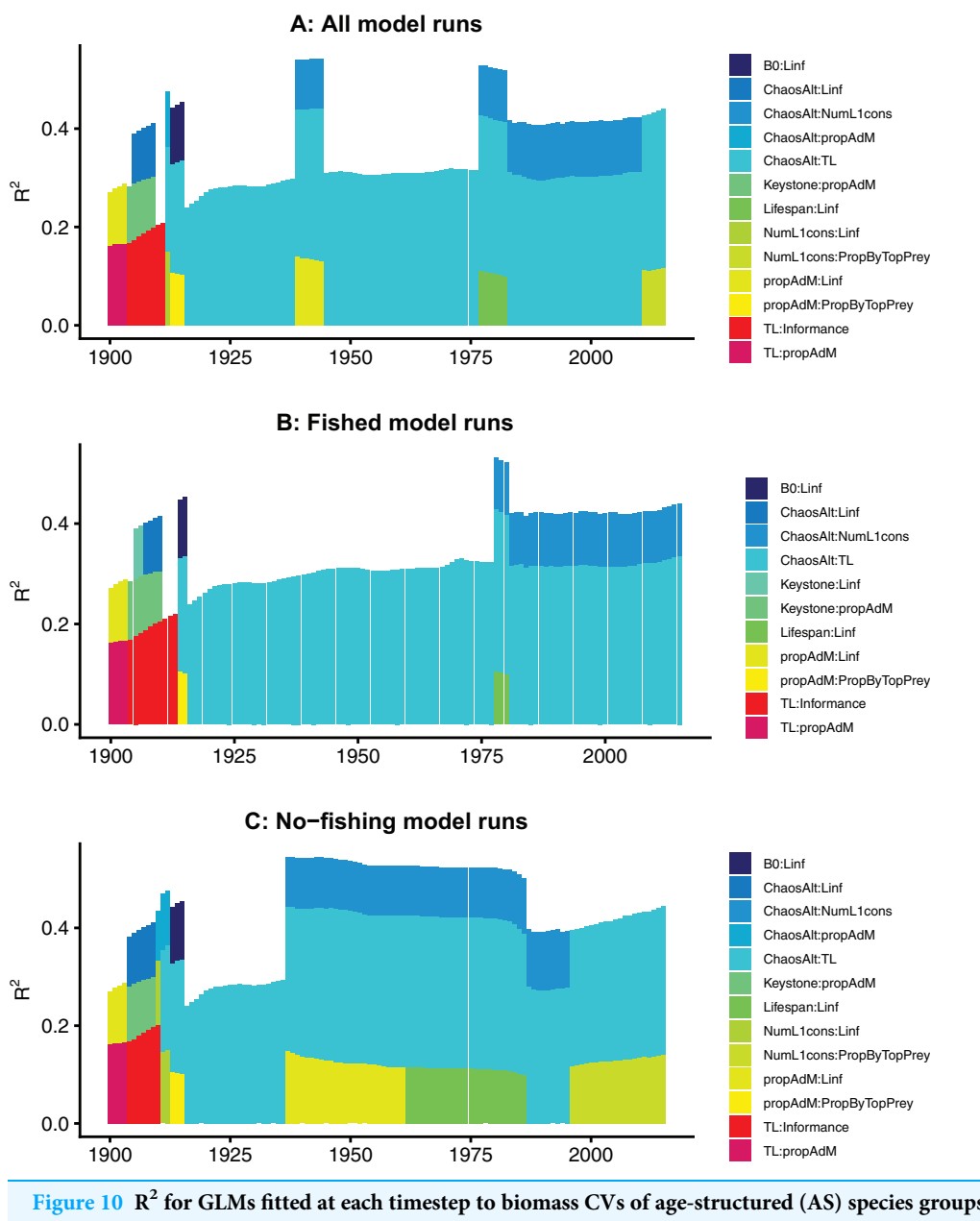

**Figure 10** $R^2$ for GLMs fitted at each timestep to biomass CVs of age-structured (AS) species groups that resulted from perturbing the initial conditions. All model runs (A), only model runs with fishing (B) and only models without fishing (C), with bars coloured by explanatory variable.

The Pearson's residuals generally showed no concerning patterns against fitted values or explanatory variables for the final GLMs (Figs. 11–15). One exception was the residuals with respect to $B_0$ for the BP model, which suggested decreasing errors with increasing $B_0$, and a possible outlier (Fig. 12).

Higher trophic level was found to be associated with lower biomass CVs for all models and ChaosAlts (Fig. 16). CVs were generally lower for ChaosAlt 'A' and 'B', which were the model runs with all initial conditions shifted up or down and by the same scalar within each run. ChaosAlt 'C' and 'D', with initial conditions perturbed based on species

**Table 7 Explanatory variables selected and corresponding r² values for GLMs fitted to ALL (all species groups) model CVs, BP (biomass-pool species groups) only model CVs, and AS (age-structured species groups) only model CVs, using model outputs from 1910–2015, with fished and unfished versions for AS.**

| Model | ChaosAlt:TL | PrimCons: $B_0$ | ChaosAlt:PrimCons | PrimCons:Inf | Total $r^2$ |
|---|---|---|---|---|---|
| ALL | 0.47 | | | | 0.47 |
| BP | 0.38 | 0.53 | | | 0.53 |
| AS | 0.32 | | 0.44 | | 0.44 |
| AS (fished) | 0.33 | | 0.45 | | 0.45 |
| AS (unfished) | 0.31 | | 0.44 | 0.54 | 0.54 |

Note:
ChaosAlt, the set of runs, grouped by method for perturbing initial conditions and whether fishing was included or not; TL, trophic level; PrimCons, number of primary trophic connections; B0, virgin biomass; Inf, informance.

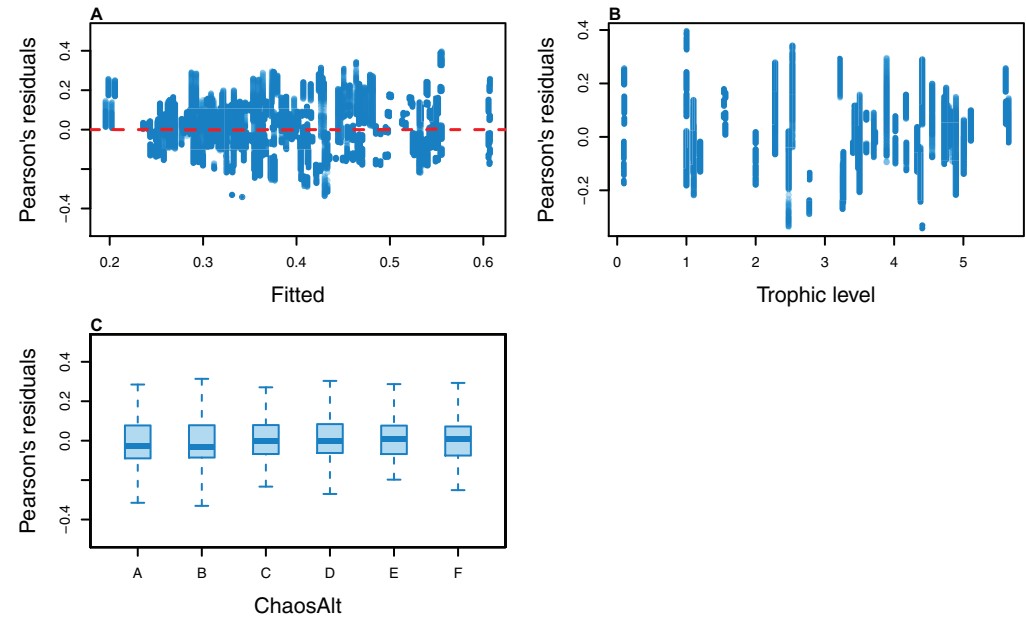

**Figure 11 Pearson's residuals for models fitted to biomass CVs of all (ALL) species groups that resulted from perturbing the initial conditions, using all model runs.** Plotted against fitted values (A), TL (trophic level) (B) and ChaosAlt (C).

group uncertainty, generally had slightly higher CVs across trophic levels (Fig. 16). This effect was also apparent in the interaction with primary connections in the AS model (Fig. 17). Biomass CVs were found to decrease with increased $B_0$ and with increasing number of primary connections for biomass pool species group (Fig. 17). The number of primary connections had the opposite effect for age-structured species groups, with more primary connections correlated with larger biomass CVs, although these CV effects were smaller (max. 11%) than for biomass pool species groups (max. 18%) (Fig. 17).

# DISCUSSION

Analysing sensitivities to initial conditions is an important part of developing complex models (*Rabier et al., 1996*; *Rosati, Miyakoda & Gudgel, 1997*; *Payne et al., 2015*;

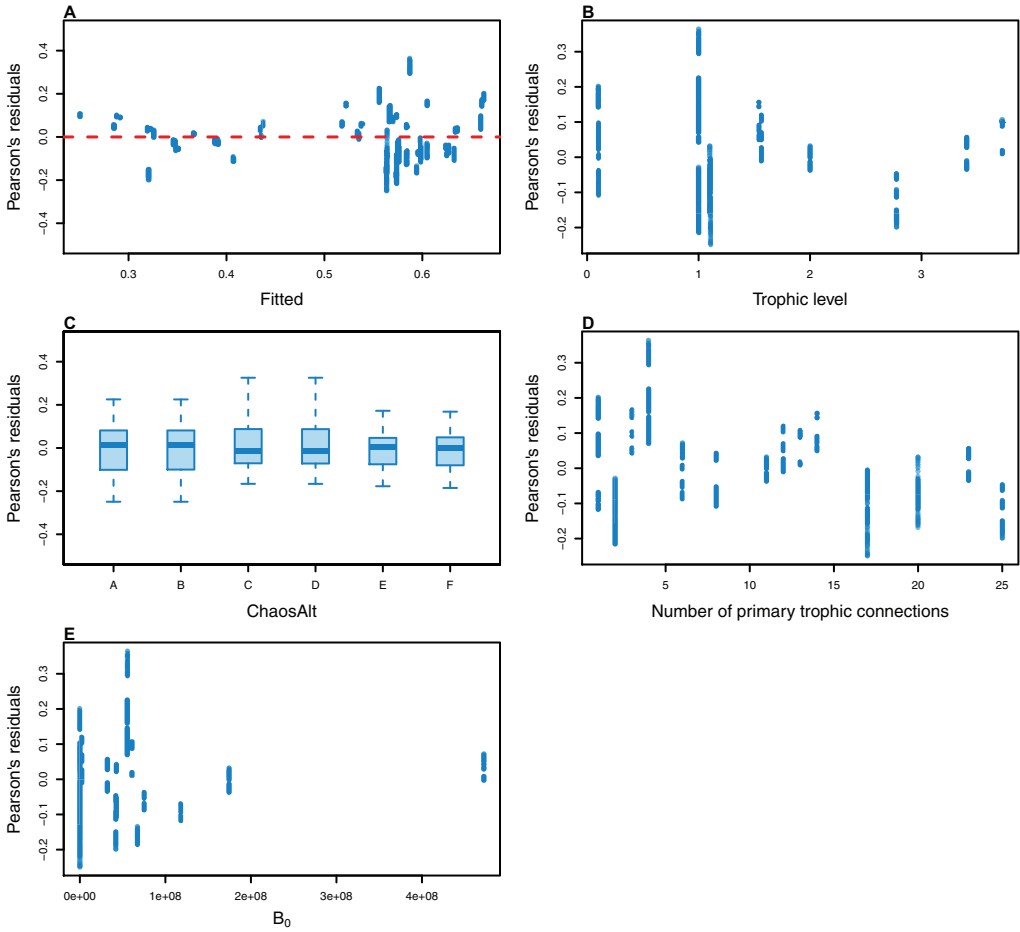

**Figure 12 Pearson's residuals for models fitted to biomass CVs of biomass-pool (BP) species groups that resulted from perturbing the initial conditions.** Plotted against fitted values (A), TL (trophic level) (B), ChaosAlt (C), Number of primary trophic connections (D) and B0 (virgin biomass) (E).

*Cheung et al., 2016*). If small perturbations to the initial conditions produce vastly different results, this may make interpreting results from the model challenging. Accounting for model uncertainty provides an envelope of model results, which tells us about the range of plausible outcomes rather than one possible instance. It is when the envelope is so wide that no result can be ascertained that it can be frustratingly un-useful, and it is important we are aware when this is the case. For example if scenarios exploring reduced fishing effort improved the general state of the ecosystem in some model runs, and deteriorated it in others, with all runs equally plausible, then we would be left none the wiser. It would be misleading to present results of only a subset or even a singular model run that does not adequately reflect the range of plausible outcomes.

We found the Chatham Rise Atlantis model was robust to initialisation uncertainty at the system level, in that we could perturb the initial conditions by small, and even quite large (up to 50%) changes, and the model produced very similar results with respect to ecosystem indicators. While the values of ecosystem indicators did retain some variability

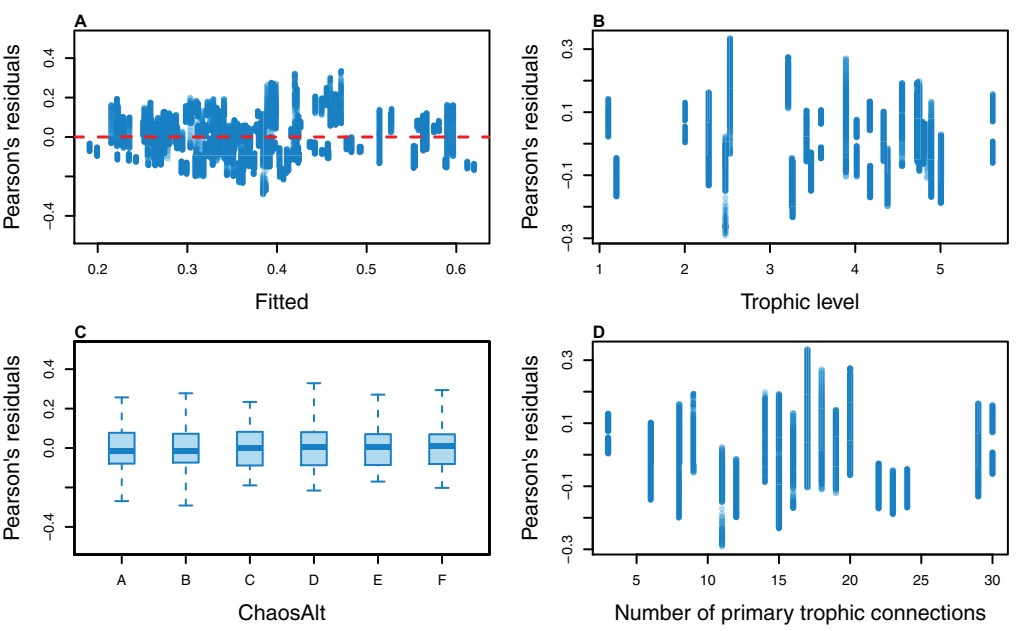

**Figure 13 Pearson's residuals for models fitted to biomass CVs of age-structured (AS) species groups that resulted from perturbing the initial conditions, using model runs with and without fishing.** Plotted against fitted values (A), TL (trophic level) (B), ChaosAlt (C) and Number of primary trophic connections (D).

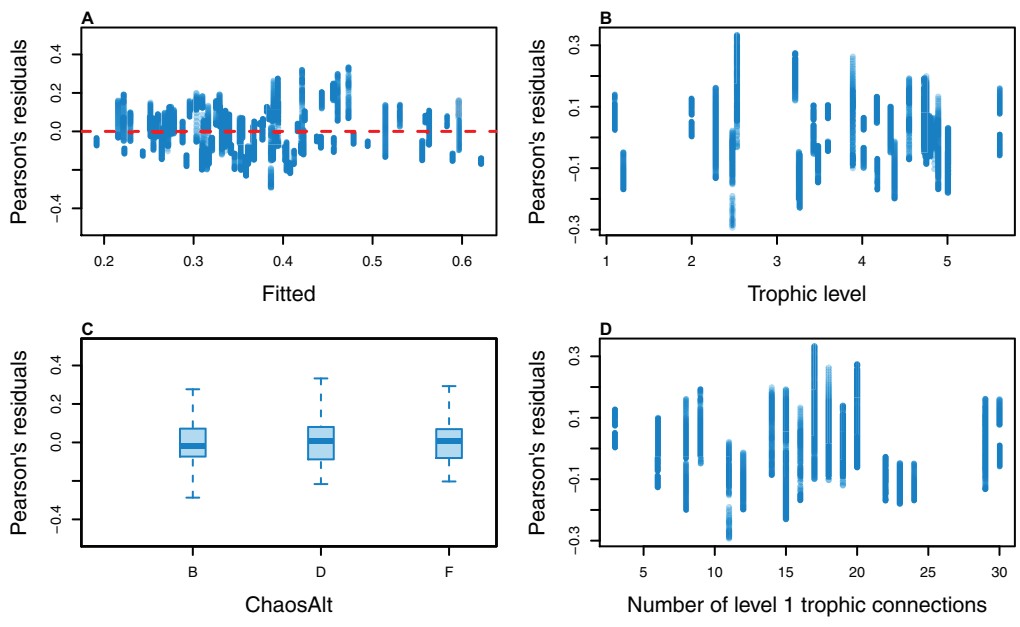

**Figure 14 Pearson's residuals for models fitted at each timestep to biomass CVs of age-structured (AS) species groups that resulted from perturbing the initial conditions, using only model runs with fishing.** Plotted against fitted values (A), TL (trophic level) (B), ChaosAlt (C) and Number of primary trophic connections (D).

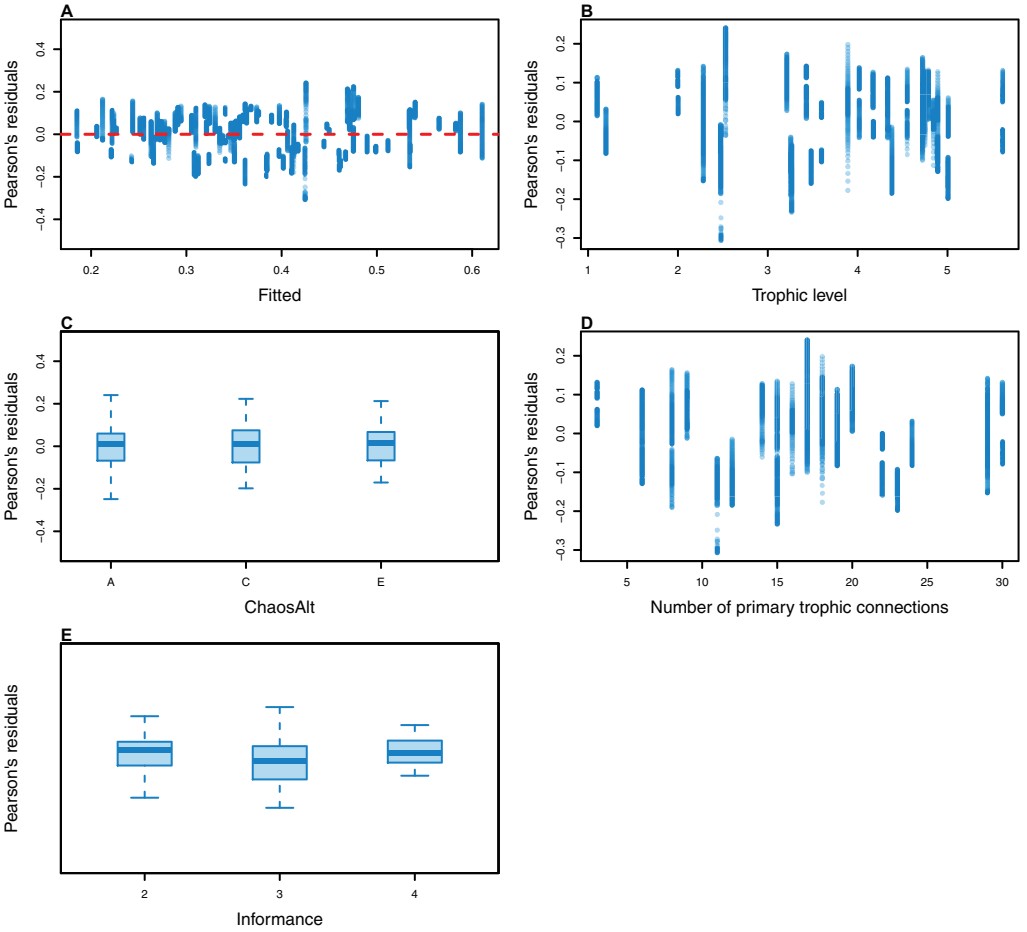

**Figure 15 Pearson's residuals for models fitted to biomass CVs of age-structured (AS) species groups that resulted from perturbing the initial conditions, using un-fished model runs.** Plotted against fitted values (A), TL (trophic level) (B), ChaosAlt (C) and Number of primary trophic connections (D) and informance (E). Informance levels 1–4 where (1) 'Poorly specified' (gold); (2) 'Some data gaps and/or poor performance' (magenta); (3) 'Slight data gaps and/or poor performance' (blue); (4) 'No data gaps, performed well, abundance index available' (green) (defined in *McGregor et al. (2019b)*). Informance level '1' did not feature in the results as these data were dropped due to 'NA' values for other explanatory variables.                                   

between model runs, the response to fishing was consistent, suggesting overall system dynamics were consistent under perturbed initial conditions. This puts us in a position to simulate scenarios using the Chatham Rise Atlantis model, including uncertainty of the initial conditions, and obtain an envelope of results with which to analyse and understand the likely responses of the Chatham Rise ecosystem.

While the system as a whole generally agreed within the range of results produced, the biomasses of some species groups varied between model runs more than others. The dynamics of some species groups appeared hyperstable as they promptly converged, while others retained variability between the runs, and for some the variability increased. We found the species groups that were more likely to have high biomass CVs were those of lower trophic levels. In nature, we expect to see more variability in the abundances
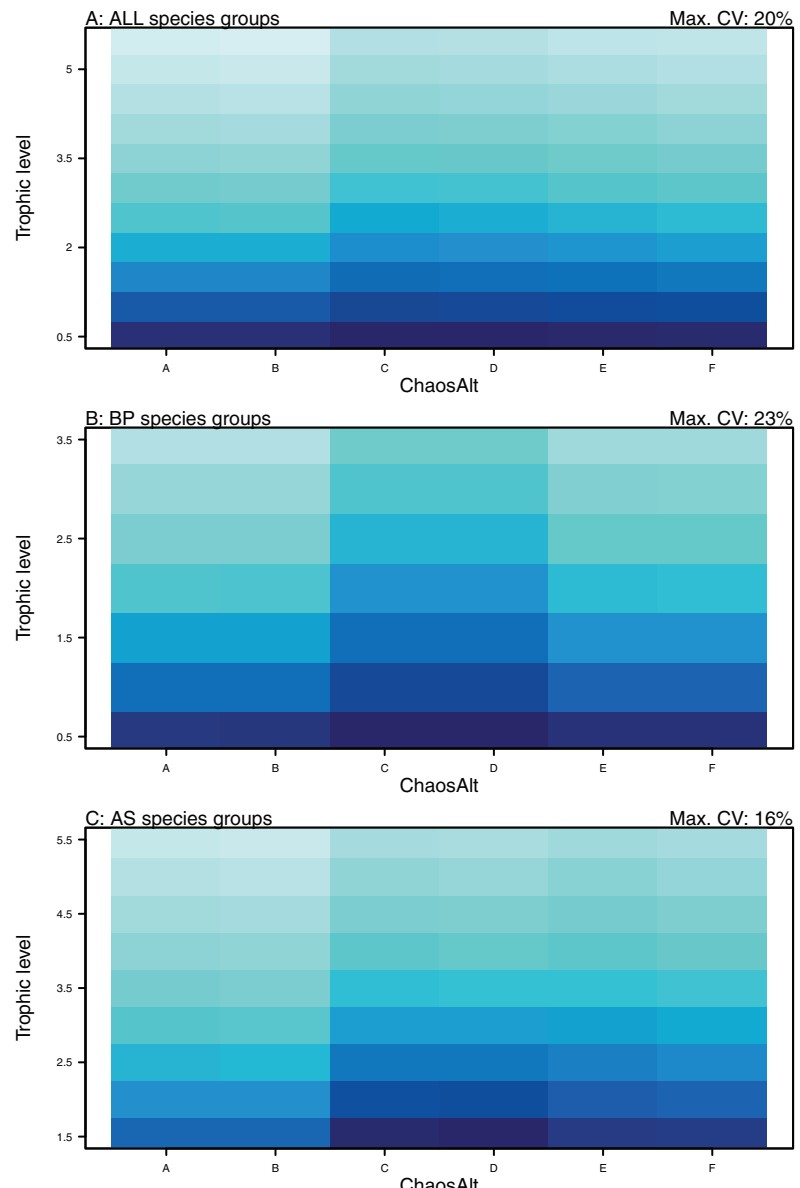

**Figure 16 GLM effects for interaction terms.** ChaosAlt:TL for ALL species groups (A), BP only species groups (B), and AS only species groups (C). ChaosAlt 'A' and 'B' perturbed all initial conditions by the same scalar for each run; ChaosAlt 'C' and 'D' perturbed initial conditions by uncertainty; ChaosAlt 'E' and 'F' perturbed initial conditions by keystoneness; ChaosAlt 'A', 'C', 'E' did not include fishing; ChaosAlt 'B', 'D', 'F' included fishing. Shading indicates the additional CV expected for each value of the interaction, with the darkest shading in each plot corresponding to the Max. CV (%) given in the top-right corner of the plot.               

of lower trophic level species, but most relevant field experts would likely suggest those patterns derive from variability within the environment (*Dippner, Kornilovs & Sidrevics, 2000*; *Dippner et al., 2001*; *Molinero et al., 2008*), which we are not applying in this study. If we combined varying the initial conditions with bootstrapping of the oceanographic

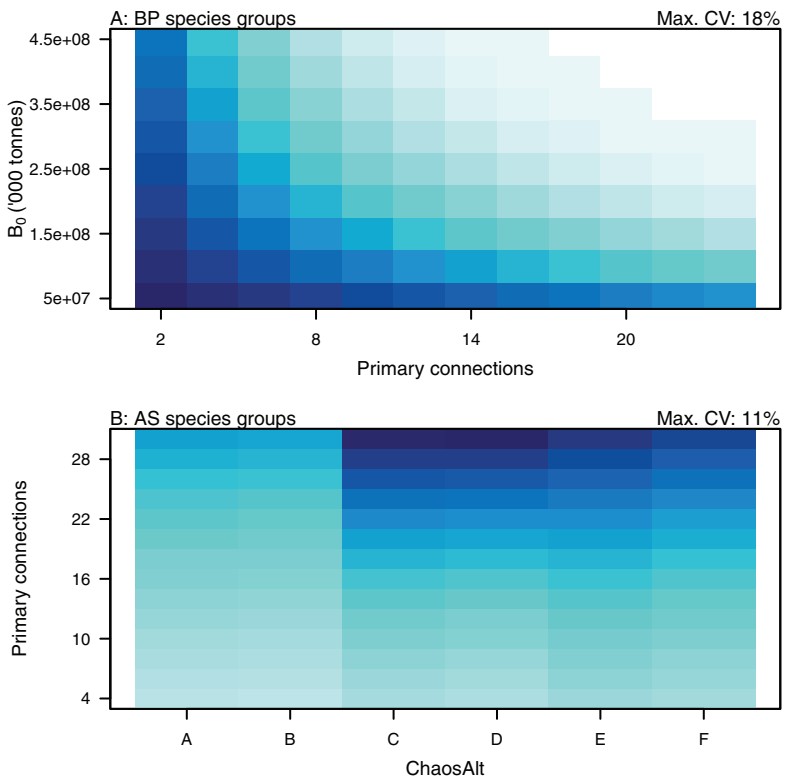

**Figure 17 GLM effects for interaction terms.** PrimCons:B0 for BP only species groups (A), ChaosAlt: PrimCons for AS only species groups (B). ChaosAlt 'A' and 'B' perturbed all initial conditions by the same scalar for each run; ChaosAlt 'C' and 'D' perturbed initial conditions by uncertainty; ChaosAlt 'E' and 'F' perturbed initial conditions by keystoneness; ChaosAlt 'A', 'C', 'E' did not include fishing; ChaosAlt 'B', 'D', 'F' included fishing. Shading indicates the additional CV expected for each value of the interaction, with the darkest shading in each plot corresponding to the Max. CV (%) given in the top-right corner of the plot.

variables, as carried out in *McGregor et al. (2019b)*, we would likely see even greater variability in the lower trophic levels.

Another aspect of the trophic level effect on variability is the way in which we have modelled the species groups in the Chatham Rise Atlantis model. First, we have the difference between species modelled as biomass-pools and those modelled with age-structure. Biomass-pool representations are more dynamic as there is little/no delay structure built in—growth is pooled across its many forms (reproductive, somatic and otherwise), so can effectively occur instantaneously, unlike in age-structured groups where maturity may take years and specific events like spawning are restrained. Given biomass-pool groups are also generally lower trophic level (with naturally higher levels of productivity and turnover), the GLM fitted to CVs of all species groups could pick up trophic level as an explanatory variable that also accounts for this group structure. Within the age-structured species groups, trophic level could also be confounded with the proportion of additional mortality. The additional forced mortality would likely be a stabilising attribute, and the proportions applied were greater for the higher trophic level

species, as these were the ones with less predation mortality in the system. That the stabilising aspect filters down through the trophic levels, with the lower trophic levels retaining variability, could suggest the extent to which this is a top-down controlled system.

The method used to perturb the initial conditions was found to be important in explaining the between-model run variability. The runs based on keystoneness did not result in the highest CVs, even though these runs perturbed the initial conditions of the species groups expected to have the greatest impact on the rest of the system. While *Paine (1969)* suggested keystone species have a stabilising effect on a system, it was more recently suggested to be more complicated than that (*Mills, Soulé & Doak, 1993*). In this study, the possible stabilising effect of keystoneness could be due to additional mortality applied to some of the high keystone species, and hence exerting a stabilising effect on the system. The runs perturbed based on uncertainty produced the greatest CVs. The effects of other explanatory variables, such as higher CVs for lower trophic levels, were consistent regardless of the method used to perturb the initial conditions. Hence, the method was not influential in how the system responded, only in how strongly it responded. It is possible the latter difference would diminish with a greater number of runs simulated for each set. In future simulations, perturbing the initial conditions based on uncertainty would seem appropriate, and should encompass the variability we would expect to see from other methods of perturbation.

One of the age-structured species groups that was most sensitive to initial conditions was the invert comm scav group (primarily scampi). When we account for uncertainty from initial conditions, the response of this group to heavy fishing is inconclusive. The heavy fishing on the system from the mid-1970s (*Ministry for Primary Industries, 2017*), some of which was targeted on scampi (*Tuck, 2016*), could easily be positive or negative for scampi based on these model results, and CVs for this species group remained high at just over 20%. In the base Chatham Rise Atlantis model (*McGregor et al., 2019b*), scampi were shown to respond to fishing in a very similar way to the fisheries stock assessment estimated biomass. The results here illustrate that the base model result for this species group, while convincing as it matched the fisheries models so well, was not representative of the many plausible results using this ecosystem model.

The effects of uncertainty from the oceanographic variables explored in *McGregor et al. (2019b)* had a greater range than those from perturbing the initial conditions. Similarly to the initial conditions uncertainty, the species most affected were also lower trophic levels. Diatoms had the highest CV at 79%, followed by carnivorous zooplankton CV at 46%. The effects of uncertainty from specification of the spawning stock recruitment relationship as it was applied to the small pelagic fish species group were explored in *McGregor, Fulton & Dunn (2019a)*. These effects were seen right through to the ecosystem indicators, and would likely be more evident if more than one species group were directly affected in the study. Further work exploring the effects of parameter uncertainty ought to be carried out with the Chatham Rise Atlantis model, and these effects compared to those from the initial conditions, oceanographic variables and spawning stock recruitment already explored. One key area of parameters to explore is the predator/prey interactions,
including the feeding functional response form and parameters. The effects of these will likely be noticed more as the model is used and hence taken away from its calibrated balanced state, where we might expect to see more variability in prey abundance.

An aspect not explored in this study is the effect of initialisation uncertainty on the model realised diets. There is plenty of scope for Atlantis diets to vary as they are the result of spatial and temporal overlap, gape-size limits, growth rates, feeding functional response, availability of other prey, predation from other predators, habitat refuge and prey preferences. As ecosystem models are generally developed to help understand flow-on effects within a system, understanding the effect of uncertainties on the species interactions could be important, and we recommend future work considers this aspect.

In the quest to provide meaningful and realistic results to simulations explored using complex ecosystem models, with high levels of uncertainty, we need to produce result envelopes, not single trajectories. It is important we move in the direction of simulating many instances of the model that account for its uncertainties, to understand how likely a given response is, and avoid presenting what may be errant or non-representative results. We know there is uncertainty in defining initial conditions of ecosystem models, so varying the initial conditions to reflect this uncertainty in model results is crucial. It is not the only area of uncertainty; there are many. Given the complexity of these models, exploring all possible uncertainties explicitly is unlikely to be tractable. It may be possible, however, to address subsets of uncertainty that encompass the broader range of the uncertainty of the model by targeting its key dynamics. The key dynamics of an ecosystem model generally consist of growth, recruitment, mortality, trophic connections, environmental effects, and initial state. Three of these (growth, mortality and trophic connections) relate directly to predation and consumption, and we could vary the feeding response function to explore the effects of uncertainties in these dynamics. Initial conditions were the topic of this study, and uncertainty from environmental effects were explored through bootstrapping the oceanographic variables in *McGregor et al. (2019b)*. This leaves recruitment/productivity, for which we could vary the spawning stock recruitment parameters. The specifics of varying these will vary between models and systems, but accounting for uncertainty with respect to four main categories: (1) initial conditions; (2) environmental; (3) feeding functional response; (4) productivity/ recruitment, is likely to cover the broad range for most systems and models.

## CONCLUSIONS

The analyses presented here provided methods for testing and understanding the effects of initialisation uncertainty of an end-to-end ecosystem model. We present results of applying these methods to the Chatham Rise Atlantis model; an end-to-end ecosystem model of a deep-sea marine ecosystem of commercial importance since the mid-1970s. We found the lower trophic species groups to be more susceptible to initialisation uncertainty. We recommend results presented from ecosystem models are presented as envelopes of plausible or likely results that reflect uncertainty such as that from the initial conditions. This will help reduce the reporting of potentially errant results that can arise from single-run simulations that do not account for any uncertainty.

## APPENDIX

Biomass trajectories (Appendix A).

## ACKNOWLEDGEMENTS

We thank Ian Tuck (NIWA), the project leader for this work.

### Funding

This work was funded under NIWA project FIFI1901. The funders had no role in study design, data collection and analysis, decision to publish, or preparation of the manuscript.

### Grant Disclosures

The following grant information was disclosed by the authors:
NIWA project: FIFI1901.

### Competing Interests

Vidette L. McGregor and Matthew R. Dunn are employed by National Institute of Water and Atmospheric Research (NIWA) Ltd. Elizabeth A. Fulton is employed by the Commonwealth Scientific and Industrial Research Organisation (CSIRO).

### Author Contributions

- Vidette L McGregor conceived and designed the experiments, performed the experiments, analysed the data, prepared figures and/or tables, authored or reviewed drafts of the paper, and approved the final draft.
- Elizabeth A Fulton conceived and designed the experiments, authored or reviewed drafts of the paper, and approved the final draft.
- Matthew R Dunn conceived and designed the experiments, authored or reviewed drafts of the paper, and approved the final draft.

### Data Availability

Data is available at GitHub: https://github.com/mcgregorv/CRAM_chaos/.

### Supplemental Information

Supplemental information for this article can be found online at http://dx.doi.org/10.7717/peerj.9254#supplemental-information.

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
