# Peer review of "Addressing initialisation uncertainty for end-to-end ecosystem models: application to the Chatham Rise Atlantis model"

_PeerJ, doi:10.7717/peerj.9254_

## Round 0.1 · original submission · Major Revisions

Please take note of comments from both reviewers.

Reviewer 1 ·

Basic reporting

A more rigorous approach to defining and testing for chaos and stability is needed. Please see attached document for specific advice.

Experimental design

Generally well designed and carried out, some specifics early in the methods and intro would go a long ways to improving the MS

Validity of the findings

As presented, findings are reasonable and sound.

Additional comments

I appreciate the amount of work that went into this effort, my comments are generally related to clarity. I also suggest a judicious amount be moved to supplementary material.

Annotated reviews are not available for download in order to protect the identity of reviewers who chose to remain anonymous.

·

Basic reporting

The manuscript uses clear and unambiguous language throughout most of the paper. Adequate background is provided. Figures and table are adequate. Largely this is a well-written manuscript; however, the structure could be improved and some references were missing.
My main concern is about the organization of the manuscript. The title led me to believe that the primary focus of the manuscript would be about approaches to exploring sensitivity to initial conditions of an ecosystem model, i.e., approaches applied to a specific model that could be applied to other similar models. The Abstract followed this logic – general information on sensitivity to initial conditions and the problems that might cause for ecosystem models, information on the specific model being used, and then recommendations on using perturbation analysis for Atlantis ecosystem models. The Discussion also followed this order – general information, specific information, and general recommendations. However, the Introduction dives right into specifics of the Chatham Rise Ecosystem, followed by general information on model stability and sensitivity to initial conditions, then reverts back to specific information on the Chatham Rise Atlantis model, and finally fairly specific recommendations about the Chatham Rise Ecosystem.
The Introduction left me wondering whether this paper is supposed to be about a broadly applicable demonstration of approaches for exploring model stability or just specific approach that can only be used for this particular Chatham Rise model. If the former is the authors’ intention, I recommend restructuring the Introduction and including a bit more general background on Atlantis models.
I also noted several missing reference citations (they just showed up as “?” in the pdf I had access to). These reference concerns and other comments are included in an annotated pdf that I submitted.

Experimental design

The work presented in the manuscript is within the Aims and Scope of the journal. The research questions are fairly clear, but see concerns noted in the Basic Reporting section of the review. The methods are rigorous and relatively clear, some additional information is needed and noted in an annotated pdf of the manuscript.

Validity of the findings

The authors have interpreted their results in a reasonable manner.

---

## Round 0.2 · Minor Revisions

A phrase like "We pave the way for addressing uncertainty from the initial conditions in complex end-to-end ecosystem models." is too broad for a case study.

Watch citations eg "Table 2: List of species functional groups modelled as biomass-pools for CRAM McGregor et al. (2019b)."and "Sensitivity of the norwegian and barents sea atlantis...."

I would like to see a section of comparative discussion about the relative effects of uncertainty in the initial conditions and uncertainty in parameter values, which are the more traditional subject of exercises like these.

Also see the comments of R1

Reviewer 1 ·

Basic reporting

Much better, but still some changes to be made to improve readability and flow of the paper, see attached suggestions.

Experimental design

Sound and better explained in this version

Validity of the findings

Unchanged from previous version, sound.

Annotated reviews are not available for download in order to protect the identity of reviewers who chose to remain anonymous.

---

## Round 0.3 · Minor Revisions

The literature cited section needs work - proper nouns are not capitalized, e.g. barents, guam.

I would still like to see a section of comparative discussion about the relative effects of uncertainty in the initial conditions and uncertainty in parameter values, which are the more traditional subject of exercises like these. Did I miss it?

The acknowledgments section is empty.

---

## Round 0.4 · accepted · Accept

Thanks for your speedy corrections.